Inter-rater and intra-rater reliability of isotonic exercise monitoring device for measuring active knee extension

Limsakul Chonnanid 1
Sengchuai Kiattisak 2
Duangsoithong Rakkrit 2
Jindapetch Nattha 2
Jaruenpunyasak Jermphiphut 3 jjermphi@medicine.psu.ac.th
1 Department of Rehabilitation Medicine, Faculty of Medicine, Prince of Songkla University , Hat Yai, Songkhla , Thailand
2 Department of Electrical Engineering, Faculty of Engineering, Prince of Songkla University , Hat Yai, Songkhla , Thailand
3 Department of Biomedical Sciences and Biomedical Engineering, Faculty of Medicine, Prince of Songkla University , Hat Yai, Songkhla , Thailand
Kashoo Faizan
Electronic publication date: 2023 Jan 16
Publication date: 2023
Volume: 11
Electronic Location ID: e14672
Received 2022 May 6; Accepted 2022 Dec 11
Copyright: © 2023 Limsakul et al.
Copyright year: 2023
Copyright holder: Limsakul et al.
License: This is an open access article distributed under the terms of the Creative Commons Attribution License, which permits unrestricted use, distribution, reproduction and adaptation in any medium and for any purpose provided that it is properly attributed. For attribution, the original author(s), title, publication source (PeerJ) and either DOI or URL of the article must be cited.
License URL: https://creativecommons.org/licenses/by/4.0/

Keywords: Isotonic exercise device, Knee extension, Inter-rater reliability, Intra-rater reliability, Knee rehabilitation, Surface electromyography, Range of motion

Funding: Faculty of Medicine Research Foundation, Prince of Songkla University and Broadcasting and Telecommunications Research and Development Fund for Public Interest (BTFP) This work was supported by the Faculty of Medicine Research Foundation, Prince of Songkla University and Broadcasting and Telecommunications Research and Development Fund for Public Interest (BTFP). There was no additional external funding received for this study. The funders had no role in study design, data collection and analysis, decision to publish, or preparation of the manuscript.

==============================
Background

The goal of this study was to assess the reliability of electromyography and range of motion measurements obtained using a knee exercise monitoring system. This device was developed to collect data on knee exercise activities.

Methods

Twenty healthy individuals performed isotonic quadriceps exercises in this study. The vastus medialis surface electromyography (sEMG) and range of motion (ROM) of the knee were recorded during the exercise using the isotonic knee exercise monitoring device, the Mobi6-6b, and a video camera system. Each subject underwent a second measuring session at least 24 h after the first session. To determine reliability, the intraclass correlation coefficients (ICCs) and standard error of measurement (SEM) at the 95% confidence interval were calculated, and a Bland–Altman analysis was performed.

Results

For inter-rater reliability, the ICCs of the mean absolute value (MAV) and root mean square (RMS) of sEMG were 0.73 (0.49, 0.86) and 0.79 (0.61, 0.89), respectively. ROM had an ICC of 0.93 (0.02, 0.98). The intra-rater reliability of the MAV of the sEMG was 0.89 (0.71, 0.96) and the intra-rater reliability of RMS of the sEMG was 0.88 (0.70, 0.95). The ROM between days had an intra-rater reliability of 0.82 (0.54, 0.93). The Bland–Altman analysis demonstrated no systematic bias in the MAV and RMS of sEMG, but revealed a small, systematic bias in ROM (−0.8311 degrees).

Conclusion

For sEMG and range of motion measures, the isotonic knee exercise monitoring equipment revealed moderate to excellent inter- and intra-rater agreement. However, the confidence interval of ROM inter-rater reliability was quite large, indicating a small agreement bias; hence, the isotonic knee exercise monitor may not be suitable for measuring ROM. This isotonic knee exercise monitor could detect and collect information on a patient’s exercise activity for the benefit of healthcare providers.

Introduction

Globally, the elderly population is predicted to grow from over 700 million in 2019 to over 1.5 billion in 2050 (UN, 2019). This group is afflicted by a variety of degenerative diseases, most notably osteoarthritis (OA). The knee joint is one of the most frequently affected locations of OA in this population (Cui et al., 2020), impacting approximately 650 million people worldwide (CDC, 2020). Osteoarthritis of the knee (OA knee) can severely limit daily activities and can cause disability since it directly affects walking ability (Liu et al., 2017). Therapeutic exercise is a key component of treatment for OA of the knee (Kolasinski et al., 2020; Esselman & Lacerte, 1994; Wiles et al., 2018; Kim et al., 2015). Research indicates that strengthening the quadriceps muscles might help alleviate discomfort and restore function to the knee joint (Anwer & Alghadir, 2014; Hislop et al., 2020; Zeng et al., 2021). The vastus medialis oblique (VMO) (Fink et al., 2007; Hart et al., 2012) is the most frequently atrophying quadriceps muscle. As a result, strengthening the VMO (Zeng et al., 2021) may help to slow the course of knee OA. Numerous guidelines advocate performing quadriceps exercises with proper technique at least 3–4 times per week (Kwok, Paton & Haddad, 2015) with a total of 10–15 repetitions each day (Vincent & Vincent, 2012).

During the COVID-19 pandemic, the amount of hospital-based therapeutic exercises significantly decreased (Boldrini, Bernetti & Fiore, 2020) because of social distancing and the hospital safety policies in place. Some hospitals switched to providing therapeutic exercise via telemedicine (Yu et al., 2020), but treatment opportunities of any type are still lacking in hospitals. Home-based exercise is a practical way for patients to perform exercises themselves, and it is more sustainable than hospital-based exercise. Current knee OA recommendations also include home-based therapeutic exercises. Knee surface electromyography (sEMG), knee range of motion (ROM), and the force, power, and speed of the knee exercises performed should all be observed to monitor knee OA patients (Heywood et al., 2019).

According to a prior study (Sengchuai & Jindapetch, 2020), the isotonic exercise monitoring device (shown in Fig. 1) monitors knee sEMG and ROM during isotonic exercise, recording and tracking the progression of each patient on a web-based platform. This device allows patients to perform the exercises by themselves, and healthcare providers can simply read the metrics of the exercise without having to observe them being performed at all. During isotonic exercise, the sEMG (Vigotsky et al., 2018) indicates muscle contraction and the ROM (Hancock, Hepworth & Wembridge, 2018) measures the movement of the knee. With these two metrics, healthcare providers are able to determine the isotonic exercise progression of each patient.

Figure 1 Isotonic exercise monitoring device and its peripheral equipment: (A) sandbag, (B) isotonic exercise monitoring device, (C) triaxial accelerometer channel for knee angle measurement, (D) sEMG channel, (E) graphic user interface (GUI).

Before using the isotonic exercise monitor in clinical applications, we need to determine if the device is qualified to measure isotonic exercise with good reproducibility and validation. Therefore, our study aims to validate this device’s functions (sEMG and ROM measurement) with the current standard instruments. To verify these two functions, we performed a comparative analysis of sEMG measurements of the right VMO using the isotonic exercise monitoring device and a Mobi6-6b wireless sEMG instrument (Phinyomark, Phukpattaranont & Limsakul, 2012; Thiamchoo & Phukpattaranont, 2022). The ROM of knee flexion and extension was also measured using the isotonic exercise monitoring device and the computer vision method (Duangsoithong, Jaruenpunyasak & Garcia, 2019). The experiment was conducted using a Con-Trex dynamometer for continual knee movement (Con-Trex, Maple Grove, MN, USA; Ema et al., 2018). The inter-rater and intra-rater reliability of the sEMG parameters and ROM measurements were compared by statistical analysis using intraclass correlation coefficients (ICCs; Koo & Li, 2016; Remigio et al., 2017) and by a Bland–Altman analysis (Giavarina, 2015).

Overall, this study’s objective was to use the isotonic exercise monitor to record the knee sEMG and ROM of healthy volunteers and compare the results to the measurements recorded using the current standard measuring devices.

Materials and Methods

Subjects

Healthy volunteers were recruited to participate in the study. Eligible subjects were at least eighteen years of age, had full ROM in both knees and had no history of musculoskeletal or neurological problems affecting either leg. The research team advertised the details of the study at Prince of Songkla University, and all eligible subjects who contacted the research team were recruited to the study. All subjects cooperated well and provided written informed consent. The study protocol was approved by the institutional ethics committee of the Faculty of Medicine, Prince of Songkla University, and was in accordance with the Declaration of Helsinki (REC 60-284-11-1).

Study design

This was an experimental study. To determine the inter-rater reliability of the isotonic exercise monitoring device, sEMG and ROM measurements were obtained and compared with standard instruments: the Mobi6-6b wireless system and the computer vision method, respectively. For intra-rater reliability, the subjects were tested one day apart using the same protocol. In both experiments, the speed, time, and force of the exercises were controlled by the Con-Trex. All measurement devices were synchronized to the time alignment of the Con-Trex to accurately determine the period of knee extension and flexion. Experienced physical therapists monitored the experiment and were in charge of the Con-Trex to ensure both safety and reliability.

Standard instruments

We used three main instruments in this study to verify the results of the isotonic exercise monitoring device. The Con-Trex dynamometer, used in this experiment to control the exercises, is popular among exercise monitoring devices because it can monitor isokinetic, isometric, and isotonic activities. Previous studies have shown it has excellent reliability for knee exercises (Maffiuletti et al., 2007) with an ICC of 0.99 for knee extension and an ICC of 0.78–0.81 for knee flexion.

We used the Mobi6-6b wireless system, a wireless device with multiple channels and applications, to measure the sEMG in this study. A previous study (Sae Jong, de Herrera & Phukpattaranont, 2021) revealed that, when given a variety of sEMG measures for facial muscle signal detection, such as mean absolute value (MAV), root mean square (RMS), and frequency domain, the Mobi6-6b had 98% accuracy in syllable recognition.

We used computer vision as the standard for measuring knee range of motion. There are numerous applications employing computer vision for monitoring human activities such as sitting posture (Kulikajevas, Maskeliunas & Damaševičius, 2021), human gait (Yu et al., 2021), and foot recognition (Duangsoithong, Jaruenpunyasak & Garcia, 2019). One study showed that applying deep learning through a convolutional neural network (CNN) gave a high, 98% accuracy for recognizing the region of interest (ROI) of the foot in images, which could then be employed to monitor the foot during knee exercises.

Experimental procedures

The methods of this experiment consisted of two main steps: preparation and exercise, as shown in Fig. 2. To prepare for the isotonic knee exercises, each subject completed a series of warm-up exercises followed by 5 min of brisk walking (Kim et al., 2014) and then a 5-min rest period. After the rest period, the subject was seated on the Con-Trex station in a comfortable posture, and his/her back was strapped to the back rest for safety. The subject’s knee position was set to 90 degrees of flexion. Because of limited space and the distance between the camera and the subject, we tested only the subject’s right leg. Next, the lever arm of the Con-Trex was attached to the ankle. The skin beyond the VMO muscle was then cleaned with an alcohol pad to remove dirt and oil. This procedure additionally reduced electrode impedance mismatch (Toledo-Peral et al., 2018). The team of researchers (J.J. and K.S.) then attached the two sEMG electrodes to the skin just past the VMO muscle with the isotonic exercise monitoring device. Specifically, the two active electrodes were attached on the line between the anterior superior iliac spine and the joint space in front of the anterior border of the medial ligament, following SENIAM guidelines (De Luca, 1997; Hermens et al., 2000). The distance between the two electrodes was 20 mm. One reference electrode was placed on the C7 spinous process. Following electrode placement, the triaxial accelerometer of the isotonic exercise monitor was strapped around the right ankle below the edge of the Con-Trex’s lever arm to measure knee ROM. Finally, the researchers configured the parameters of the isotonic exercise monitoring device as shown in Table 1. The researchers also set up the video camera system 2 m away from the subject with a lateral view perpendicular to the subject.

Figure 2 Experimental settings of the: (A) Con-Trex’s lever arm, (B) isotonic exercise monitoring device, (C) Con-Trex device, (D) anterior superior iliac spines, (E1 and E2) electrode placement (E1: 80% of the distance between (D) and (F), E2: 20 mm next to E1).

Table 1 Configuration of each device in this experiment.

Device	Measurement	Configuration	Outcome measure	
Con-Trex	N/A	Isotonic knee flexion/extension exercise with 25% of 1-RM	N/A	
Mobi6-6b	sEMG	One channel with one pair of electrodes for sEMG (sampling rate: 1,024 Hz) Amplifier: 1,000-fold	MAV RMS	
Canon EOS 550D with computer vision method	Video	Digital camera mounted on a tripod with 1 m height and 2 m away with a perpendicular orientation (sampling rate: 30 frames per second) Resolution: 18 megapixels 1,920 × 1,080 pixels and the flash disabled.	ROM	
Isotonic exercise monitor	sEMG ROM	Two channels with one pair of electrodes for sEMG (sampling rate: 1,000 Hz) and triaxial accelerometer on the ankle (sampling rate: 100 Hz) Amplifier: 1,500–fold	MAV RMS ROM	

For the exercise procedure, the one-repetition maximum (1-RM) of each subject was measured by having each subject perform just one isotonic knee extension exercise, followed by a 5-min rest period (Freitas de Salles et al., 2009; Lin et al., 2008). The subject then performed two isotonic knee exercise repetitions at 25% of his/her 1-RM to become familiar with the measurement and to check the sEMG signal and ROM measurements from the isotonic exercise monitoring device, followed by another 5-min rest period. The subject then performed six isotonic knee exercise repetitions using the Con-Trex device, with the isotonic exercise monitor recording the sEMG and ROM of the knee, and the computer vision program also measuring the ROM of the knee. Following this, the subject rested for another 5 min (Freitas de Salles et al., 2009; Lin et al., 2008), during which the researchers (J.J. and K.S.) replaced the sEMG cable of the Mobi6-6b device’s electrodes with those from the prior experiment, configuring the Mobi6-6b device parameters as illustrated in Table 1. The subject then performed six isotonic knee exercise repetitions using the Con-Trex device, with the Mobi6-6b device recording the sEMG signals. At the end of the experiment, the sEMG electrodes on the skin were removed, and the location of the electrode attachment was marked for the next experiment for each subject. All subjects returned 24 h later and performed the same experiment to evaluate the intra-rater reliability of the isotonic exercise monitoring device compared with standard protocols. All data were automatically collected by all instruments used in the experiment.

Data analysis

The motor unit action potential (MUAP) is the combination of all muscle fiber action potentials from a single motor unit (Raez, Hussain & Mohd-Yasin, 2006). In this study, the MUAP was detected by a skin surface electrode positioned on the VMO muscle. At the electrode, the raw sEMG was detected and amplified. Differential amplifiers are often utilized as first stage amplifiers. Before collection, low- or high-frequency noises and other signal noises were filtered from the signal as demonstrated in Fig. 3.

Figure 3 Feature extraction process of sEMG in: (A) the isotonic exercise monitoring device and (B) the Mobi6-6b device.

The raw sEMG signal from the isotonic exercise monitoring device was first amplified 1,500-fold and then filtered by the bandpass filter circuits at 10–500 Hz. The notch filter circuits subsequently reduced the 50 Hz as powerline noise (Akwei-Sekyere, 2015) in the sEMG. Next, this sEMG was sampled at a frequency of 1,000 Hz as part of the software used to gather data into the computer. The signal was also filtered with an infinite impulse response (IIR) bandpass filter at 10–500 Hz and then normalized to a standard value by dividing the maximal peak of sEMG during muscle contraction, according to the methods of prior studies (Biviá-Roig, Lisón & Sánchez-Zuriaga, 2019; Gagnat, Brændvik & Roeleveld, 2020). Time synchronization by Con-Trex was then used to partition this sEMG by knee extension periods. Finally, the MAV and RMS were extracted from the sEMG signal and calculated using the following two equations:

(1) MAV=1N∑i=1N|Xi|

(2) RMS=1N∑i=1NXi2

Here, MAV is the mean absolute value of the sEMG signal, N is the number of sEMG samples, Xi is the sEMG amplitude at the i index, and RMS is the root mean square of the sEMG signal. The raw sEMG signal from the Mobi6-6b was amplified 1,000-fold and then transferred using Bluetooth technology. In the subsequent procedure, the computer received wireless sEMG data with a sample rate of 1,024 Hz, which was then filtered with an IIR bandpass filter at 10–500 Hz. Finally, the sEMG features were extracted using the same methods that were used with the isotonic exercise monitoring device.

To calculate ROM, as illustrated in Fig. 4, a triaxial accelerometer ( ±2 g range) of the isotonic exercise monitor was strapped around the right ankle of the subject and then sampled at 100 Hz. The roll ( ϕa) and pitch ( θa) angles (Narkhede et al., 2021; Franco et al., 2021) were then calculated using the following equations:

Figure 4 ROM calculation process in: (A) the isotonic exercise monitoring device and (B) the computer vision method.

(3) ϕa=arctan⁡(ay/az)

(4) θa=arctan⁡(−axaysin⁡ϕ+azcos⁡ϕ)

Here, ax, ay, and az are represented linear accelerations determined along three orthogonal axes, respectively.

The ROM was measured from the starting angle of knee flexion to the final angle of knee extension based on the time-synchronization of Con-Trex recordings. During knee exercise, the subject was recorded using a digital camera (Canon EOS 550D) as part of the computer vision approach. The digital camera was also mounted on a tripod that was 1 m height and perpendicular to the subject. There are six primary ROM calculation processes based on convolution neural networks in object detection (Hakim & Fadhil, 2021). First, an image of the knee at its initial angle of flexion was cropped from the top left to the bottom right using sliding windows to find the located foot ROI in the image. According to the methods outlined in a previous study (Duangsoithong, Jaruenpunyasak & Garcia, 2019), a pre-trained convolutional neural networks (CNNs) model was then used to classify each cropped image to determine whether it contained a foot or not. If this CNNs model predicted the that image contained a foot, the ROI was stored in memory until the end of the last sliding cropped image. The algorithm then picked a single ROI from several overlapping entities, a process known as non-maxima suppression (Rothe, Guillaumin & Van Gool, 2015). Next, the centroid of the foot ROI and the fixed location of the knee joint was labelled by the team researcher (J.J.). In this experiment, the knee joint of the subject was fitted into the Con-Trex lever arm’s rotating point. The location of the camera and the Con-Trex both remained the same. The angle of the knee joint along the Y-axis was calculated using the ROI of the foot and the location of the knee joint, as shown in Eq. (5) and illustrated in Fig. 5. Then, the final knee extension degree was noted, as labeled by the Con-Trex. The ROM was then determined by subtracting the end degree of knee extension from the beginning angle of knee flexion. After calculating the range of motion, the video of the participant’s exercise was deleted immediately. Only the final ROM calculation of the participant was stored on the computer.

Figure 5 Example of knee joint angle calculation using the trigonometry rules in Eq. (5) where θ is the angle of the knee joint.

(5) θ=arctan⁡(A/B)

Here, θ is the angle of the knee joint, A is the distance on the X-axis between the centroid of the foot ROI and the location of the knee joint, and B is the distance on the Y-axis between the centroid of the foot ROI and the location of the knee joint.

Statistical analysis

For the sample size calculation, we used the ICC based on two raters (k = 2) (Jerez-Mayorga et al., 2020), and the expected effect size to assume differences in EMG and ROM measurement in an isotonic exercise monitoring device compared with standard protocols was computed (Koo & Li, 2016). We estimated the power (1 − β) by defining the expected reliability (ICC, ρ1) as 0.6, the significance level ( α) as 0.05, and a power of 0.90 (Bujang & Baharum, 2017). The sample size estimate indicated that at least 20 subjects were required.

The baseline characteristics of participants were presented as means and standard deviations (SD). The mean value of both devices on each day was calculated to evaluate the inter-rater and intra-rater reliability using ICCs. Moreover, an absolute reliability was calculated with the standard error of measurement (SEM). The research team proposed a method for interpreting ICCs, which has been used in many published studies (Koo & Li, 2016). An ICC value higher than 0.90 indicates excellent reliability. ICC values between 0.75–0.90 and 0.50–0.75 indicate good and moderate reliability, respectively. Lastly, ICCs less than 0.5 indicate poor reliability. The Bland−Altman plot analysis was conducted to demonstrate the agreement between the isotonic exercise monitor and the standard measurement devices. Systematic bias was determined using the 95% CI of mean difference (Gerke, 2020). If zero is included in the 95% CI, no systematic bias may be inferred. A narrow 95% CI (LoA95) indicates more stability. We utilized all analyses using the R Program (Vienna, Austria) version 4.0.3.

Results

This study enrolled a total of 24 subjects. However, four subjects were removed because the sEMG signal could not be detected; in one example, the patient’s subcutaneous tissue was thick, obscuring the sEMG measurement. Table 2 summarizes the characteristics of the final twenty study participants, stratified by gender.

Table 2 Summarizes the characteristics of the study participants, stratified by gender.

Characteristics	All subjects (N = 20)	Males (N = 11)	Females (N = 9)	
BW ( kg) mean ± SD	65.27 ± 13.39	68.59 ± 10.68	61.21 ± 15.81	
BH ( cm) mean ± SD	167.24 ± 10.11	173.77 ± 5.47	159.26 ± 7.89	
BMI ( kg/m2) mean ± SD	23.42 ± 4.61	22.73 ± 3.50	24.06 ± 5.64	
Age ( yrs) mean ± SD	30.10 ± 6.96	30.64 ± 6.44	29.44 ± 7.89	
Note:

BW, body weight in kilogram ( kg) units; BH, body height in centimetre ( cm) units; BMI, body mass index in kg/m2 units; yrs, years; and SD, standard deviation.

The results of our experiments also show that the isotonic exercise monitoring device could be used to monitor knee exercises under supervision. The mean values of the sEMG parameters and ROM are shown in Table 3; the average MAV and RMS were 0.17 and 0.21, respectively. The mean knee ROM value was 87.18±1.79 degrees, which was similar to the knee ROM values of healthy volunteers (Saranya, Poonguzhali & Saraswathy, 2019).

Table 3 Results of inter-rater and intra-rater reliability of the isotonic exercise monitor.

	Mean (SD)	Inter-rater ICCs (95% IC)	SEM1	Intra-rater ICCs (95% IC)	SEM2	
MAV	0.17 (0.04)	0.73 [0.49–0.86]	0.02	0.89 [0.71–0.96]	0.01	
RMS	0.21 (0.03)	0.79 [0.61–0.89]	0.01	0.88 [0.70–0.95]	0.01	
ROM	87.18 (1.79)	0.93 [0.02–0.98]	0.47	0.82 [0.54–0.93]	0.76	
Note:

ICCs, intraclass correlation coefficients; SEM1, standard error of measurement of Inter-rater ICCs (95% IC); SEM2, standard error of measurement of Intra-rater ICCs (95% IC); MAV, mean absolute value of sEMG signal; RMS, root mean square of sEMG signal; and ROM, range of motion of knee joint.

The inter-rater ICCs of the MAV and RMS between the isotonic exercise monitor and the Mobi6-6b were 0.73 (95% CI [0.49–0.86]) and 0.79 (95% CI [0.61–0.89]), respectively. These results show moderate and good reliability of sEMG measurements between the two devices. Furthermore, the between-day intra-rater ICCs of the MAV and RMS of the isotonic exercise device were 0.89 (95% CI [0.71–0.96]) and 0.88 (95% CI [0.70–0.95]), respectively, indicating the sEMG data had good reliability between the two testing days (Koo & Li, 2016).

The inter-rater ICCs of ROM between the isotonic exercise monitor and the computer vision method was 0.93 (95% CI [0.02–0.98]), indicating excellent reliability. The intra-rater reliability of the ROM between days was 0.82 (95% CI [0.54–0.93]), illustrating good reliability of the device (Koo & Li, 2016).

The Bland–Altman plot compared two experimental measurements in three parameters: MAV, RMS, and ROM. At first, the mean difference of MAV between the isotonic exercise monitoring device and the Mobi6-6b was 0.0058 (95% CI [−0.0027 to 0.0143]) (Fig. 6). The width of LoA95 was −0.0463 to 0.0580. Figure 7 shows the RMS parameter. The mean difference between the isotonic exercise monitoring device and the Mobi6-6b was 0.0017 (95% CI [−0.0054 to 0.0088]). The width of LoA95 ranged from −0.0419 to 0.0452. Lastly, the Bland–Altman analysis indicated that the mean ROM difference between the isotonic exercise monitoring device and the computer vision method was −0.8311 (95% CI [−0.9000 to −0.7622]). The width of LoA95 was −1.2533 to −0.4089, as shown in Fig. 8.

Figure 6 The Bland–Altman plot of the inter-rater agreement of MAV between the isotonic exercise monitoring device and the Mobi6-6b (n = 40).

Figure 7 The Bland–Altman plot of inter-rater agreement of RMS values between the isotonic exercise monitoring device and the Mobi6-6b (n = 40).

Figure 8 The Bland–Altman plot of inter-rater agreement of ROM values between the isotonic exercise monitoring device and the computer vision method (n = 40).

Discussion

This isotonic exercise monitoring device tested in this study was developed in a laboratory for the purpose of monitoring knee exercises. As required by sEMG instrument standards, this study established that this device may be used in healthy volunteers for real-time exercise monitoring without major consequences, such as serious joint or muscle injury (Tankisi et al., 2020). The MAV and RMS of the isotonic exercise monitoring device revealed moderate to good inter-rater reliability when compared to the Mobi6-6b device. Additionally, the results demonstrated the excellent reliability of the ROM calculated using the isotonic exercise monitor compared to ROM values acquired using the computer vision method. In terms of between-day intra-rater reliability, the sEMG and ROM measurements were shown to have good reliability. Moreover, the Bland–Altman plot of the MAV and RMS showed no systematic bias (Gerke, 2020). Because our results show that the tested isotonic exercise monitor accurately records standard sEMG measurements, which are beneficial for signal recognition in high-precision measurements, categorization of limb position, and analysis of motion movement (Sae Jong, de Herrera & Phukpattaranont, 2021; Thiamchoo & Phukpattaranont, 2022), we conclude that the isotonic exercise monitoring device and its parameters could be used in clinical practice as a tool for monitoring knee rehabilitation. For example, it might be used to monitor a training station in a general primary care unit. Additionally, this isotonic exercise monitor offers objective data monitoring, real-time biofeedback, and automatic data recording as shown in Fig. 1.

While the ROM’s inter-rater reliability was excellent, its confidence interval was large (0.02, 0.98; Thangjai, Niwitpong & Niwitpong, 2020; Hazra, 2017) due to the angle’s scale being highly sensitive. In this experiment, the computer vision algorithm also detected the centroid of the foot image while calculating knee range of motion, instead of using the ankle’s lateral malleolus (Hancock, Hepworth & Wembridge, 2018). However, the Bland–Altman plot pointed out a small, −0.8311 degrees bias in the ROM measurement, with a narrow 95% CI. According to a previous study (Guzik et al., 2020), a typical minimal clinical important differences (MCID) of knee ROM values is about 6.81 degrees. The tested isotonic exercise monitor may be good for monitoring exercise, but is not suited for use as a high-precision measuring instrument.

Numerous exercise devices combining robotic and computer systems have been created in recent years (Hu, Wang & Wu, 2021; Zhang & Liu, 2021). Advances in information processing have also substantially improved treatment evaluation options. Using existing exercise devices, we can observe the progression of significant exercise parameters such as force and range of motion. The Con-Trex is a standard exercise monitor, which has been shown to be reliable in several previous studies (Maffiuletti et al., 2007; Ema et al., 2018). In accordance with the Con-Trex, the isotonic exercise monitor was able to analyze objective data that might assist in determining exercise activities.

There are some limitations of the study. No satisfaction questionnaires were administered to the study participants prior to or following the use of this isotonic exercise monitor. The sample size of this study was also rather small.

In future work, this monitoring device could be reduced in size to accommodate mobile monitoring applications. Furthermore, additional research on specific subsets of patients, including older adults, should be done to determine the device’s efficacy in rehabilitation programs.

Conclusion

This study confirms the reliability of an isotonic exercise monitoring system for knee rehabilitation. The results show that the sEMG parameters and ROM calculated with this device have moderate to excellent inter-rater reliability with the results of standard monitoring devices. The tested monitoring device also illustrated good intra-rater reliability between the two study days. The Bland–Altman analysis showed there was no systematic bias of MAV and RMS values, but did show a small bias in ROM values between the isotonic exercise monitoring device and the computer vision results. Therefore, this isotonic exercise monitor could not be used as a high-precision knee ROM measuring device, but may be used to track a patient’s exercise and to report the patient’s exercise metrics to healthcare providers. For future research, this device should be made smaller and interfaced with mobile phones for home-based exercise monitoring.

Supplemental Information

Supplemental Information 1 Dataset and code for inter-rater reliabilty and intra-rater reliabilty.

Click here for additional data file.

We express our gratitude to the Department of Epidemiology and Dr. Ply Chichareon from the Department of Internal Medicine, Faculty of Medicine, Prince of Songkla University for their support in statistical analysis.

Additional Information and Declarations

Competing Interests

Author Contributions

Human Ethics

Patent Disclosures

Data Availability

The authors have no conflicts of interest to declare.

Chonnanid Limsakul conceived and designed the experiments, performed the experiments, analyzed the data, prepared figures and/or tables, authored or reviewed drafts of the article, and approved the final draft.

Kiattisak Sengchuai conceived and designed the experiments, performed the experiments, analyzed the data, prepared figures and/or tables, and approved the final draft.

Rakkrit Duangsoithong conceived and designed the experiments, prepared figures and/or tables, and approved the final draft.

Nattha Jindapetch conceived and designed the experiments, prepared figures and/or tables, and approved the final draft.

Jermphiphut Jaruenpunyasak conceived and designed the experiments, performed the experiments, analyzed the data, prepared figures and/or tables, authored or reviewed drafts of the article, and approved the final draft.

The following information was supplied relating to ethical approvals (i.e., approving body and any reference numbers):

The Ethical Committee of Faculty of Medicine, Prince of Songkla University approved the study to be carried out within its facilities (Ethical Application Ref: 60-284-11-1).

The following patent dependencies were disclosed by the authors:

The isotonic exercise monitoring device was licensed with the Department of Intellectual Property (DIP), Thailand (No. 16629).

The following information was supplied regarding data availability:

The code and raw data are available in the Supplemental Files.

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
