# Peer review of "Inter-rater and intra-rater reliability of isotonic exercise monitoring device for measuring active knee extension"

_PeerJ, doi:10.7717/peerj.14672_

## Round 0.1 · original submission · Major Revisions

There are several parts of the manuscript that needs to be improved. The introduction needs justification, precise details of the literature, and more description of how the EMG signals were processed. Authors also need to improve the discussion in terms of their own interpretation of the results in light of the existing literature.

Please follow the format of the journal such as citation and introduction.

Please get the manuscript editing by a person proficient in scientific writing.

Please answers the reviewer's comments and make amendments to the manuscript.

Reviewer 1 ·

Basic reporting

No comment

Experimental design

The methods need significantly more detail. Below are questions that the authors should try to address
How were subjects recruited? Where were they recruited from? How many possible subjects were recruited?
How was subject size determined?
What camera was used to record the video? What was the height of the camera?
Were any filtering techniques applied to the sEMG data? What was the sampling frequency?
Why did the authors not use Bland Altman technique to quantify the agreement between the two analyses?
You can just write out Table 3 in your methodology instead of having another table.

Validity of the findings

I have several questions/comments regarding the results
In the written results please state that you're reporting 95% CIs

I know you mention the high CI and SEM of the ROM due to their sensitivity, I think a Bland-Altman may be very valuable for ROM

Reviewer 2 ·

Basic reporting

The manuscript describes the experiment and post-analysis to check the reliability of an EMG system by the development of several test with volunteers in two different sessions.
The article conforms to professional standards of expression, but must be improve in some written parts to improve it technically and clear the ambiguous text.
The background context provided in the article include sufficient information to sketch the work and explain the field of knowledge.
The structure of the article should be modified in some sections. Figures and tables must be close to the main text and appropriately labelled. Also, some figures are missed and could help the reader to understand your work.

Experimental design

Research questions are not well defined.
The method must be described in more detail and information about the equipment used.

Validity of the findings

The impact is not demonstrated because of the lack of information. Authors must consider to underline the data and make a robust presentation of the findings.

Additional comments

Find additional comments attached.

Annotated reviews are not available for download in order to protect the identity of reviewers who chose to remain anonymous.

Reviewer 3 ·

Basic reporting

-The authors should proofread the entire paper for grammar.
-The article structure did not fully follow traditional IMRD format with a literature review section as part of the introduction.
-Literature review section should be part of the introduction instead of its own section.
-Table 1 – please also report descriptives for male and female subjects in addition to the total
-Experimental procedures should be written in paragraph format and consider a schematic to illustrate the procedures.
-Discussion is very brief and the reliability of measures needs to be discussed in context of other equipment. Some of which is brought up in the introduction.

Experimental design

Greater detail is needed regarding processing of the raw data. Specifically the filtering of the sEMG signals because if this was not done appropriately the results cannot be evaluated. Please see SENIAM guidelines as well as work by Deluca to ensure this was done correctly and properly reported.

Validity of the findings

Cannot evaluate until the data processing is more clearly explained.

Additional comments

Introduction, Paragraph 1 – Please add another sentence describing loss of function due to knee OA in elderly population. That would improve the transition to paragraph 2.

Line 37 – need space after population

Currently paragraphs 1 and 2 could be merged to a single paragraph.

Table 2 –‘Result’ should be labeled ‘Outcome Measure’

Experimental procedures should be written in paragraph format and consider a schematic to illustrate the procedures.

Typically EMG signals are filtered prior to computing values such as RMS, MAV, etc. Please provide justification as to why this was not performed. It is not evident whether appropriate processing of the EMG data was conducted and as a result makes it challenging to evaluate the rest of the paper.

Statistical analysis needs greater description.

Discussion is very brief.

---

## Round 0.2 · Major Revisions

In addition to any outstanding reviewer comments, please also make the following changes:

Line 73, change to the experiment was conducted.

Line 78, what do you mean by "complication"?

Line 78, Please rewrite your purpose example: " The purpose of your study was to record the EMG signals and ROM of the knee and to compare the recording with a standard reliable device."

Line 82. The sentence to describe the method is not needed.

In lines 124–144, you are describing the procedure you adapted to record EMG signals. Please explain why you chose to record the monitoring isotonic exercise device first and then Mobi6-6. Please provide a rationale for the procedure, or have you adapted this procedure from research conducted earlier? Additionally, if the procedure was adapted from previous research, how did you manage fatigue?

Please add another picture of the placement of the electrodes.

In line 177, 179, 180, 182... what is ROI? Please provide the full form of ROI initially.

Please add the reliability and validity of Con-Trex, Mobi6-6b, and Canon EOS 550D with computer vision method in the method section before the procedure and justify their selection as standard reference.

In line 229, were there any minor complications? What do you mean by no major complications? It would be better if author used "no adverse events" instead of "complications."

The Bland Altman plot represents a greater number of data points than the actual (n=20). The number of dots in the plot needs to be 20. Please correct the image.

Line 234., please rephrase the sentence for better understanding.

Line 233-243, please include studies to support your results about the reliability of monitoring isotonic exercise devices.

Line 245-252, excellent explanation of the large confidence interval with reference.

Line 253 and 254. The sentence requires references.

Lastly, the manuscript looks good but needs further English editing to reduce passive and complex sentences.

Reviewer 1 ·

Basic reporting

The authors have addressed all of my concerns.

Experimental design

No comment

Validity of the findings

No comment

Additional comments

Thank you for responding to my feedback.

Reviewer 3 ·

Basic reporting

Thank you for the revisions.

Experimental design

Thank you for the revisions. I would ask the authors provide some detail on the software used to perform statistical analyses.

Validity of the findings

Thank you for the revisions.

Additional comments

Thank you for the major revisions to the manuscript.

---

## Round 0.3 · Minor Revisions

Dear Authors,
The manuscript looks much better now. i have only one comment before a possible acceptance of this manuscript.

Please add a statistical section to the article before the results in a separate paragraph.

Statistical Analysis section. In particular, sample size, use of appropriate statistical measures, correction for multiple testing, effect size, degrees of freedom, and reporting of relevant figures such as the statistic, n, and exact p-values/confidence intervals as needed.

Regards

---

## Round 0.4 · Minor Revisions

Dear Authors,

The manuscript still needs minor work before I recommend it for publication.
The changes are indicated in the annotations in the PDF file attached.
Please refer to my earlier suggestions that you add a statistical analysis section. It's still missing or inadequately written. Please look at the other articles on the same topic to get an idea of how to put together a good statistics section.

Please take help from PeerJ staff to revise and check the grammar of the included sections in your manuscript.

---

## Round 0.5 · accepted · Accept

Thanks a lot for addressing all the comments made on the manuscript. I am happy with the current version, and the manuscript is ready for publication.